# Listening to the Shenzhen Primary Healthcare Context to Adapt the mhGAP-IG.v2 for the Assessment of Depression: Qualitative Workshops with Primary Healthcare Leaders

**DOI:** 10.3390/ijerph19052570

**Published:** 2022-02-23

**Authors:** Kendall Searle, Grant Blashki, Ritsuko Kakuma, Hui Yang, Harry Minas

**Affiliations:** 1Global and Cultural Mental Health Unit, Centre for Mental Health, School of Population and Global Health, University of Melbourne, Melbourne, VIC 3010, Australia; h.minas@unimelb.edu.au; 2Nossal Institute for Global Health, The University of Melbourne, Melbourne, VIC 3010, Australia; gblashki@unimelb.edu.au; 3Centre for Global Mental Health, London School of Hygiene and Tropical Medicine, London WC1E 7HTE, UK; ritsuko.kakuma@lshtm.ac.uk; 4Monash Institute for Health & Clinical Education, School of Primary Health Care, Monash University, Melbourne, VIC 3168, Australia; hui.yang@monash.edu

**Keywords:** depression, mhGAP-IG.v2, primary healthcare, China, Shenzhen, conceptualisation, cultural adaption, contextualisation, depression presentation

## Abstract

In Shenzhen, despite recent primary and mental healthcare reform, Primary healthcare doctors (PHC) have limited access to diagnostic tools and a significant mental health treatment gap presides. The World Health Organization’s (WHO) mental health gap intervention guide (mhGAP-IG.v2) offers a non-specialist and evidence-based guide for the assessment of depression however requires adaptation to the context of use. Bilingual (Mandarin and English) qualitative research was undertaken with 30 PHC leaders from Shenzhen to compare their assessment approach for depression against the mhGAP-IG.v2 in order to identify context-specific modifications for a local guide. Local assessment differentiators included: a need for culturally sensitive translation of depression symptoms; a preference for a broad, non-hierarchical symptom presentation (including somatic, behavioural and anxiety items); national prioritisation of suicide patients; the integration of family into the cycle of care; limited primary care awareness of a depressive episode in Bipolar Disorder; and China’s specialist-led diagnostic approach. Contextual modification of mhGAP-IG.v2 is recommended to take account of China’s unique cultural and primary health system response to depression. Ongoing mental health training is required to develop professional confidence in the recognition of mental disorders.

## 1. Introduction

In 1980, Shenzhen, located in China’s southern province of Guangdong, became a special economic zone (SEZ) and opened the door to China’s first market-led economy [1]. Since then, it has transformed from a small fishing village into a “global technology hub and innovation metropolis” which contributes significantly to the government purse and leads the way for national socio-economic reforms [2,3]. Industrialisation precipitated mass internal migration [4,5] which has created a city with a unique population demographic: the estimated 12 million permanent residents are outnumbered by “temporary migrants”, a vast mobile workforce who access the city for employment [2,6,7]. Whilst urbanisation has brought benefits such as rising living standards and the development of public infrastructure and services [2,8], the rapid changes to life-style and exposure to emotional stressors has been paralleled by increasing rates of mental disorders [4,9,10]. With regards to depressive disorder, studies in key workers [7,11,12] and vulnerable populations [9,13,14] have identified prevalence rates significantly higher than the national life-time estimate of 7.4% [15].

Depression is a complex and heterogeneous disorder [16]. How depression is conceptualised as a condition varies according to whom you talk to, where they are from and the times they are living in [17,18]. China’s unique social-political past has significantly shaped its modern recognition of, and response to, depression in ways that are radically different to Western concepts [19,20,21]. According to the communist-inspired thinking of the late 1940′s, for instance, mental health was seen as the acquisition of “up-standing moral character” which could only be achieved through an individual’s struggle to overcome capitalism through hard labour [21]. During the decade-long tumult of the cultural revolution (1966–1976), the very disciplines that underpin modern psychiatry in the West (i.e., psychology, anthropology, sociology) were outlawed [20,21,22] and academics denounced as the nation participated in mass burnings of literature. Symptoms identified in Western diagnoses of depression were considered highly unnationalistic characteristics [20,23]:“withdrawal” was not in the spirit of collectivism; and “fatigue” was construed as “laziness” [21]. These common perceptions (or misperceptions) and associated attitudes of stigma propagated the need for emotional concealment, as a way of self-preservation [21].

The implications for the clinical picture of depression on the current generation are far-reaching. Parents brought up in a climate of emotional caution have transferred their practices onto their children, so that, the existence of depression in a family member is still hidden as part of a cultural norm today [24,25]. Low levels of mental health literacy amongst the general population [25,26,27] and even medical professionals persist [28] and widespread structural, interpersonal and intrapersonal mental health stigma severely hamper access to care [26,29,30,31]. According to a Guangzhou-based study, 92% of participants with an identifiable mental disorder suitable for treatment claimed “they did not perceive the need for treatment”. Amongst those who felt treatment was necessary, more participants reported attitudinal (83%) rather than structural (65%) factors as a barrier to accessing treatment [32].

In China, the formal diagnosis of mental disorders (including depression) is considered to be the responsibility of psychiatrists [33,34]. Previously, the locally published, *The Chinese Classification of Mental Disorders* (CCM) was their classification system of choice, however, recently, there has been an uptake in the key international classifications systems: *The International Classification of Diseases* (ICD) and *The Diagnostic and Statistical Manual* published by the World Health Organization (WHO) and the American Psychiatry Association respectively [34,35]. This change in preference has been facilitated by the organic convergence between systems along with WHO’s commitment to forging a global diagnostic language [36,37]. For example, the latest version, ICD-11 [38] has become more sensitive to the role of cultural context on disease presentation [39] and has even embedded Traditional Chinese Medicine (TCM) within its classifications [40]. Similarly, the updated CCMD-3 [41] demonstrates a long-awaited alignment with international systems regarding the once politically sensitive diagnostic label of Neurasthenia, which is now only considered after excluding all anxiety and depressive disorders [42,43].

Current global thinking is that primary care is the best place to detect and treat depression [44,45]. Apart from patients being more likely to seek medical advise during a depressive episode [46], primary care doctors are also privy to a patient’s family background and potential stressors for depression [47]. In high-income countries depression screening instruments are both available and validated for use at primary care sites [48] and doctors are well placed to provide both psychosocial interventions and drug treatment [49]. Over the recent decade, China has steadily followed suit, through its rapid operationalisation of a vast network of Community Healthcare Centres (CHC) as a first point-of-care for non-acute conditions [49,50] and intent to integrate mental healthcare into primary care [51,52,53]. However, in practice, not all CHCs have access to an in-house physician with mental health training, nor are diagnostic tools or depression protocols standardly available, leaving primary care doctors poorly equipped to recognize depression in their communities [33].

Even if CHC doctors did have access to standard diagnostic tools, their use may not be appropriate for primary care. The classification systems have their foundations in hospital-based psychiatry, where the observation of in-patients led to the identification of symptom clusters and the eventual categorisation of a condition [54]. They are driven by the need to make treatment decisions (i.e., to prescribe drug treatment or not) [54,55,56] and are not designed to detect the early symptoms and signs of depression that differ from those of acute clinical disease. Many diagnoses are outdated and were shaped without the aid of research evidence as available today [54,57]. Public health researchers, for example, suggest that it might be more appropriate to conceptualise depression as a multi-dimensional condition, where symptoms can occur in any order over time, and interact with each other to intensify or naturally stabilise an individual’s state [45,54]. Furthermore, earlier versions of ICD and DSM have been criticised for not being culturally sensitive [58].

To overcome these shortfalls and to address the global mental health treatment gap, World Health Organization (WHO) published the Mental Health Gap Intervention Guides (mhGAP and mhGAP-IG.v2), a decision-support tool to assist non-specialists working in the community with the assessment, management and follow-up of seven priority Mental, Neurological and Substance Use Disorders (MNS) (i.e., Depression; Psychosis; Epilepsy; Child and Adolescent Mental Health and Behavioural Disorders; Dementia, Disorders Due to Substance Use; Self-harm/Suicide) [59,60]. The guide was developed using a rigorous peer review process to determine the evidence-based, best-treatment practice and financially affordable options for each condition. [61,62] It is freely available in seven languages (Arabic, English, French, Italian, Marathi, Russian, Spanish (not officially available in Mandarin Chinese)) and has been used in over 90 countries as a framework to review mental health policy, for health curriculum development and training, and an assessment aid at the point of care [63,64].

mhGAP-IG.v2 is composed of seven chapters (one per priority condition). Clinical guidance for the assessment of depression is provided in the first section of the first module (pg 21–25). It does not assume the format of a screening instrument (i.e., no scoring element included) but instead provides an initial overview of the common presentations of depression, then employs a flow-chart to communicate key decision-making steps for: (i) Does the person have depression? (ii) Are there other explanations for the symptoms? Depression is considered if the person experiences at least one core mood-based symptom (i.e., persistent depressed mood and/or diminished interest in or pleasure from activities) AND several other concerns from a symptom listing for at least 2 weeks, AS WELL AS, has difficulty with daily functioning in relationship domains. Depression is only “considered likely”, after ruling-out other physical conditions that might resemble depression (e.g., anaemia, hyperthyroidism, mania and recent bereavement).

mhGAP-IG.v2 potentially offers community healthcare centres in Shenzhen a useful resource to improve the detection (and management) of depression in the community. However, the guidelines are generic in nature, are based-on Westernised classification systems and were developed without consultation with the Chinese primary care community (NB earlier consultations only involved specialist-level mental health experts in China). Guideline adaptation provides an accepted, cost- and time-effective approach for knowledge transfer from one context to another [65], however, recent learnings from implementation studies are clear: “context is everything” [66]. Thus, the objective of this research, was to compare primary care’s assessment approach for depression against the mhGAP-IG.v2 in order to identify context specific modifications that would improve the relevance and acceptability of the guide for use in Shenzhen’s primary healthcare sector.

Primary healthcare research is a relatively new field in China. So, too, are policy directives to integrate mental health within primary care. Currently, to the authors’ knowledge, there are no other published China-based studies that have focused specifically on practice-based evidence for the development of the WHO mhGAP-IG.v2 depression guidelines.

## 2. Materials and Methods

### 2.1. Fieldwork

Thirty primary care leaders from Shenzhen, organised into seven sub-groups (3–5/group), participated in a two-hour long, bilingual (Mandarin and English), audio-recorded, task-based discussion which focused on the assessment section of the depression component of mhGAP-IG.v2. These discussions constituted the first session of a two-day Mental Health Workshop which was organised for two cohorts (Cohort 1, *n* = 20; Cohort 2, *n* = 10) of delegates attending the “*Monash-Shenzhen General Practice Clinical Leadership Training Programme*” hosted by Monash University in Melbourne. Ethical approval was received to conduct this research from The University of Melbourne’s Human Ethics Advisory Group (HEAG), June 2018 (ID:1851783.1).

### 2.2. Participants

Inclusion criteria: identified as a “primary care leader” by the Shenzhen Health Capacity Building and Continuing Education Centre, (the medical regulatory body and vocational training provider for primary care qualification and certification in Shenzhen) as determined by academic excellence (i.e., examination scores for national medical examinations in the top quartile) and assessed during competitive panel interviews to receive a government funded place on The Monash-Shenzhen General Practice Clinical Leadership Training Program [67]; holding senior manager grade or above at a Community Healthcare Centre in any one of Shenzhen’s ten districts; hold clinical responsibility for patients and “see” depression patients; either hold or intend to study for the primary healthcare mental health certification; English Language proficiency

Exclusion criteria was not applied: Preliminary research indicated that doctors do not formally diagnose depression, thus criteria restricting participation to only those who diagnosed more than a predetermined number of patients with depression/week was considered inappropriate. Nor were participants excluded based on the number of years working in CHC due to the relatively new introduction of the primary healthcare sector into China’s national healthcare system.

### 2.3. Recruitment

All delegates were invited to attend a face-to-face information session in English and Mandarin (KS) where the study objectives and requirements were provided along with plain language statements in Mandarin Chinese/English and a copy of the mhGAP-IG.v2 (assessment section only). Delegates were able to consider their participation over a one-week period and text their reply to the researcher privately. Thirty of the 33 delegates agreed to participate and their written, informed consent was received. Reasons for declining included [68] attendance at an alternative specialist training and [68] doctors who did not currently work in Shenzhen. Participants were free to leave the study at any time without bias. An independent well-being co-ordinator was made available to participants throughout the course of the study should they feel the need to seek support.

### 2.4. Mental Health Workshop

The guiding principles for research were derived from the ADAPTE Manual and Resource Toolkit for Guidelines Adaptation (v2.2009), Step 15, [69,70] to explore applicability, acceptability and transferability of the depression component of the mhGAP-IG.v2 for use in community healthcare centres from the perspective of primary care physicians. Each sub-group was presented with a large poster of the assessment flow chart extracted from the mhGAP-IG.v2, Chapter on Depression, Assessment Section, Pages 21–25 [60]. The moderator asked the participants to consider each section of the flowchart (i.e., common symptoms; symptom presentation; diagnostic exclusions) in relation to their current working context by using three step-wise tasks:- a free-listing [71]; a ranking [72]; and label generation [73]. See Table 1 for further details. This task-based discussion approach was adopted because of its suitability for a workshop format—providing an engaging stimulus for discussion; effectively directing participant attention towards detailed content matter; providing an environment for reflection and group discussion; and drawing out contributions from all members of the sub-groups, including individuals who may be more reticent to contribute or prefer to give considered input after more reflection [74].

### 2.5. Moderation and Translation

The primary researcher (KS) facilitated all seven task-based discussion groups. She has been affiliated with the mental health gap action movement since its launch in 2008 [75], has studied Mandarin Chinese and worked in China. She was supported by a native-born mandarin speaker both an experienced simultaneous translator and qualitative researcher active in the field of primary healthcare (HY). All bi-lingual discussions were audio-recorded in their entirety (including the simultaneous translations provided by HY) and KS/HY collaboratively produced transcripts careful to maintain the source language of response (i.e., English and/or/both Mandarin). The Mandarin components of the transcripts were translated into English (HY) and compared against the simultaneous translations (KS) with any differences/nuances noted and discussed. Additionally, supporting workshop materials (i.e., sticky labels, comments on posters) were collated, catalogued and translated (KS) where necessary.

### 2.6. Analysis

A thematic content analysis [76] was performed (KS) using the English language transcripts to extract emergent, recurring themes (managed in Nvivo 11 software) which were considered in conjunction with the supporting visual materials produced during the workshop (e.g., comments on posters, sticky labels). Emergent themes were back-checked against original transcripts in the source language by the simultaneous translator and native-born Chinese researcher (HY) and discussed and collated into potential suggestions (from the perspective of participants) for guideline adaptation. Frequency counts (by group) were used to analyse the output for the free-listing exercise, Common Symptoms of Depression in Shenzhen.

Results were interpreted by maintaining PHC leaders’ responses as central drivers of the research process and recognizing individual authors’ unique expertise, positionality and intrinsic bias in relation to research findings. Five authors in total (including the primary researcher and the simultaneous translator) contributed to the final script. Three were Western, one Asian and one from mainland China. The primary researcher, although Western, has lived and worked in China and other Asian countries. Together they brought multiple perspectives regarding the management of mental health (i.e., specialist/GP/Community and global/local). All five had prior research and capacity building experience in both developed and non-developed countries. They jointly commented, discussed findings and collaboratively reached a consensus view on results.

## 3. Results

### 3.1. Demographics

Thirty doctors (17 females and 13 males), working in 22 different community health centres, located across seven out of 10 districts of Shenzhen, participated in this research. See Figure 1 below. As a convenience sample, no doctors from Nanshan were included due to a cohort effect at the time of the mental health workshop. There were no doctors from Pingshan, nor Dapeng, where access to this overseas training scheme has not yet been established. All participants were senior managers or directors of a community healthcare centre. Participants had an average of 9 years’ medical experience and an average of 4 years working in their current CHC of employment.

### 3.2. Primary Healthcare Leaders’ Conceptualisation of Depression

The assessment section of mhGAP-IG.v2 was developed taking into account the general classification principles of the *International Classification of Diseases* and *Diagnostic and Statistical Manual* as available at that time (i.e., ICD-10 and DSM-IV) [60]. Using this criterion, patients with a positive diagnosis for depressive disorder display the core “mood” symptoms of depression (i.e., either/or/both persistent depressed mood; diminished interest in, or pleasure from, activities) as well as any of several, additional, itemised symptoms for at least two weeks (see Figure 2, Scheme A).

When asked to draft the contents for the “common presentations” of depression according to their own experience of working in community healthcare centres, the groups generated an average of 9.6 items (ranging from 4–18 items) each. There was agreement between two or more groups in relation to 14 symptom items. (See Figure 2, Scheme B). Overall, in contrast to the mhGAP.IG.v2, the listing was composed of predominantly physical and behavioural symptoms rather than psychological symptoms.

Specifically, all of the groups excluded the mhGAP-IG.v2 core symptom of “persistent depressed mood” (Scheme A, Core symptom, Item 1) from their listing. Instead, the most widely mentioned symptom was “cease communication with other people”. This was closely followed by “fatigue/sleep problems” which is a non-core item in mhGAP-IG.v2 (Scheme A, Additional Symptom, Item 1). Doctors’ addition of overt depression-related behaviours such as “crying/weeping”, “withdrawal from activities” and “low libido” (Figure 2, Scheme B, Items 3, 5, 13), somatic symptoms such as headache, chest palpitations and non-specific pain (Figure 2, Scheme B, Items 6, 9, 11), as well as anxiety, further differentiates their listing from that of mhGAP-IG.v2.

### 3.3. Response to Decision Making Flow Chart for “Does the Person Have Depression”?

Overall, the groups wanted to see more synergy between the guide and the typical presentation of a potential depression patient in Shenzhen. Ideally, the decision-making steps would parallel and support the natural course of conversation between doctor and patient. Groups discussed the following concerns:

*A broader symptom listing to avoid screening-out on core symptoms preferred:* A recurring debate across all seven sub-groups was the value of a definitive depression diagnosis in the context of primary care practice. Many groups queried the appropriateness of excluding the possibility of depression in patients who did not display core symptoms but had additional symptoms as listed in the mhGAP-IG.v2. When thinking about their preventative role in community-based medicine, doctors discussed their obligation to provide psycho-social/supportive care for patients with additional symptoms, irrespective of the presence of core symptoms. There was a strong tendency for groups to merge the boxes for ‘Core Symptoms’ and ‘Additional Symptoms’ bringing together the range of symptoms. One group also asked to include rare symptoms. In addition, they were concerned to take into account the course of disease and, in particular, how the symptom profiles might change as depression becomes more severe.


*“Combine the core symptoms and additional symptoms. We are not psychologists. You can have depression without displaying the core symptoms. You don’t want to rule out depression on the basis of the core symptoms (i.e., be too specific). Draw the two steps together. Two steps waste time. Patients (in Shenzhen) typically present with symptoms listed as additional. A broad symptom list is more suitable for the Shenzhen context.”*
R2 (*n* = 4)

*Suicide to be elevated as a priority concern:* Patients at high risk of suicide were repeatedly mentioned as a high priority concern by several groups. These groups felt suicide ideation should be addressed much sooner in the overall guide and higher up within the symptom listings. Additionally, they sought stronger reminders for doctors to work collaboratively with the family members of potentially suicidal patients, who they considered could facilitate visits to the specialist. They expressed their concerns regarding suicide-related stigma and the role doctors can play to avoid escalation to police involvement.


*“Ask about suicide earlier in consultation, because it is more serious. Specialists should see suicide patients. However suicide patients feel ashamed. They are also afraid so they will not go to a specialist. Doctors call the police to find out where the suicide patients are. The family will take the suicide patient to the specialist. It is better to involve the family prior to the police for suicide patients. Patients only have a short time with specialists.”*
R2 (*n* = 4)


*“The ordering of the symptom list does not reflect doctors’ health priorities. The list order should place severe symptoms first. Move suicidal thoughts to the top of the list. Or even consider suicide intention first. Suicide is a symptom of depression.”*
G2 (*n* = 3)

*Emphasise that family facilitates health seeking and mental health care:* Several groups wanted to modify the decision-making flow chart in the mhGAP-IG v2 with stronger reference to the family’s role in consultations so that doctors were reminded to work with them from the outset.


*“Activate the family network in parallel with the referral to specialist. A double branch is required on the guide to show this.”*
R1 (*n* = 5)

*Introduce guidance on the use of depression screening instruments:* Some groups raised the potential for the guide to incorporate recommendations on depression screener usage (e.g., K10, PHQ9). Doctors queried the appropriate point in patient care to administer a screener (with some groups suggesting it as a first step in diagnosis); which screeners should be used; favorable settings to administer screeners; how to obtain training; and whether other health professionals, such as nurses, could also be trained to administer screeners.


*“Introduce a screening questionnaire! Nurses can support doctors by conducting depression screeners. Screening by the nurse can take place either before or after consultation with the doctor.”*
G1 (*n* = 5)

### 3.4. Response to Ruling out Alternative Diagnoses

*Checking for physical conditions that can resemble or exacerbate depression:* The positioning of this question within the flow-chart sequence (i.e., after considering mood-based symptoms) caused considerable debate and prompted doctors to reflect upon primary care’s role within the health system. Some mini-groups wanted to invert the guide to align decision-making with their current day-to-day clinical response to patients. Their preference was to initiate a range of physical tests (e.g., blood tests, auscultation etc.) and then refer onto a specialist who would assess for mood-related symptoms. They wished to distance themselves from the responsibility of assessing for mood-related symptoms. Other mini-groups, which included doctors with a specialisation in Chinese Medicine, welcomed the prioritisation of mood-related symptoms as part of their role and as a necessary preliminary step in the assessment of depressive disorder.


*“GPs treat the physical symptoms. Patients without a physical condition are sent to the specialist. Questions about psychological condition are not asked because GPs have no treatment to give. Patients with unresolved physical condition are referred to the specialist. The order of the sequence should be changed to place the consideration of physical problems first. For example, check physical cause of dizziness, chest pain first.”*
G2 (*n* = 3)

*Mania:* When asked about mania, groups tended to agree that CHC doctors did not see patients with mania (unless visiting patients directly in their homes as part of a wider, intersectoral, healthcare response). Participants explained that patients with mania are typically taken straight to the hospital-based specialists by their family and thus bypass CHCs. One group explained that mania was poorly understood by doctors, was often not differentiated from schizophrenia, and fell out of their scope, treatment skillset and CHC resource structure. Furthermore, the concern that patients presenting with a depressive episode of bipolar disorder are at risk of misdiagnosis and inappropriate antidepressant treatment (without a mood stabiliser) appeared to be new information for most doctors in both groups. Overall, they welcomed the guidance and “new” information in this area.


*“GPs do not see mania. Mania patients are not distinguished from schizophrenia. People report mania to the police! The family takes mania patients directly to the psychologist… Psychologists treat mental disease. It is difficult for GPs to treat mental disease. There are no resources to treat mental health in the CHC and doctors are not trained to treat mental disease. Doctors need further training.”*
G2 (*n* = 3)

*Grief should be acknowledged earlier in consultation:* Given their community-based role, several groups agreed it would be more natural for the guide to signpost “loss” much earlier in the assessment process as this would be more effective use of consultation time and fitted closely with their community-based role.


*“Move major loss up to somewhere near the top of the guide, so that it is early in consultation. It is easy for us to ask about bereavement.”*
R1 (*n* = 5)

## 4. Discussion

Firstly, this research highlights the importance of using culturally sensitive translations for guideline development. In keeping with previous research, doctors chose to use euphemisms when discussing depression [33]. Their (spontaneous) conceptualisation of depression was devoid of core mood related terminology such as “persistent depressed mood”. Instead, they used expressions such as, “cease communication/talking less” and “fatigue/tiredness”, which are colloquial synonyms for “sadness and depression” and “hurt or despair” respectively [20,77]. Given the backdrop of intense mental health stigma [24,25,26,29,30,31], patients commonly communicate “depressed mood” in terms of bodily complaints, perceiving these to be a more acceptable premise upon which to seek medical attention [20,78]. It might be interpreted, that doctors subconsciously take account of the cultural masking of depression and accommodate for this in their choice of language. Thus, an adapted guide would do well to further investigate these social-cultural and linguistic nuances.

This research emphases PHC leaders predisposition for a broad and less stringent symptom listing for depressive disorder which includes more somatic symptoms and behavioural expressions of depression than what is currently included in the mhGAP-IG.v2. They were uncomfortable with screening-out patients based on “core symptoms” for fear of excluding patients who may need monitoring and support. Their response is consistent with the strong preventative healthcare remit for primary care in Shenzhen [51] and resonates with the Lancet Commission on global mental health and sustainable development, which advocates for the reframing of mental disorder, to consider depression as a spectrum of symptoms, that can occur in any order, at any time, rather than a clinical category as used by psychiatry [45]. This approach promotes physician consideration of prodromal symptoms of depression and individual-specific distress [54] to facilitate early interventions and the clinical staging of care over the life course of the condition [79,80]. With these factors in mind, modifying the mhGAP-IG.v2 to include an appropriately merged and extended symptom listing is recommended.

PHC leaders included anxiety on the symptom listing for depression. Anxiety has been identified as China’s most prevalent mental disorder [15] and is frequently comorbid with depression [81] and other commonly seen conditions in primary care. Other studies suggest that worry/anxiety are on a precursory pathway to depression [82,83]). Cultural beliefs depict depression and anxiety as “one condition with two faces”. For instance, a person may be considered anxious if he talks too much, and depressed if he talks too little. Thus any guidance regarding the assessment of depression should also consider anxiety and its prevalence in the community.

PHC leaders demonstrated a heightened awareness of suicide and suggested reconfiguration of mhGAP-IG.v2 to reinforce the benefits of early detection of suicide risk factors and supportive action for their patients. Physicians are most likely aware of the highly publicised suicide events amongst employees of the Shenzhen-based electronics manufacturer, Foxconn [84]. Additionally, as part of China’s commitment to the Sustainable Development Goals to reduce the number of suicides by 30% by 2030 [85] (in 2016 the national suicide rate was as high as 7.05 per 100,000 [86]), the informal identification (i.e., not achieved through the use of an assessment instrument) of patients potentially at risk of suicide was introduced as a key performance indicator within all community healthcare clinics in Shenzhen. However, despite the overall decline in the national suicide rate, expanding populations, such as migrant workers and the elderly, are still at increased suicide risk [87]. Given Shenzhen’s particular population demographic (i.e., significantly sized, young, migrant-worker population), a stronger emphasis on the routine assessment of suicide risk factors within the depression component of mhGAP-Ig.v2, as well as, linkage with subsequent chapters designated to “Self-harm/Suicide” (pg 131–140) [60] should be considered.

The findings highlighted that PHC leaders recognised the invaluable role that family members play in facilitating access to care. This aligns with a study that reported that 86% of doctors without a mental health specialisation considered having the support of family was helpful in managing depression patients [28]. Apart from Chinese societal norms where the duty of care often falls on family members, there is increasing evidence that family-based mental health interventions, offered as part of an integrated healthcare system, improve health outcomes and potentially avoid relapse of depressive episodes [88]. Furthermore, from a national policy perspective, the activation of the family network is considered to be an essential component for the initiation (e.g., assessment) and on-going delivery of care to sufferers of poor mental health [89]. Referencing the role that family members play during the assessment process (and thereafter) within the guidelines could potentially pay respect to, and harness, a viable and cost-effective resource in comprehensive mental health care.

PHC leaders suggested including directives within the mhGAP-IG.v2 to use depression screeners to identify depressive disorder in their patients. Currently, screeners are not standardly available at CHC level in China [33] and guidelines on optimal depression screening approaches in primary care differ by country [16]. In general, screening is considered helpful (particularly in high risk populations) as a first step in depression diagnosis in healthcare settings which have:- access to reliable screening tools; resources to follow-up with diagnostic assessments; and availability of appropriate treatment [16]. As China’s primary care services develop, its health policy guidelines might consider the merits of integrating a stepped assessment approach into primary care practice and, in particular, which tools should be made widely available. The Kessler Psychological Distress Scale (K10) and the two- and nine-item versions of the Patient Health Questionnaire (PHQ-2 and PHQ-9) were mentioned. Both have been validated for use in multiple settings [90,91,92,93] and may provide CHCs with viable options in primary care due to their reliability and brevity.

This research identified a significant gap in primary care knowledge with regards to bipolar disorder. PHC leaders reported that they rarely saw patients with mania in their clinics and they considered the diagnosis and treatment of patients with bipolar affective disorder as the responsibility of the hospital system. They acknowledged that they had limited understanding about this condition and that it was commonly confused with psychotic conditions. The possibility that primary care doctors may encounter patients with bipolar disorder experiencing an episode of depression, as is regularly the case in USA [94,95], was a new consideration for them. Recently measured prevalence rates of bipolar disorder were lower in China than in Western Countries, although there is debate around how bipolar disorder is classified in the Chinese Classification of Mental Disorders (CCMD) [96]. Additionally, the health-seeking status of individuals suffering from bipolar disorder is suspected to be low due to their concerns of stigma, prior poor experience of care and high treatment expense [97]. Clearly this is an area that medical training and guidelines would need to address in the future and an expansion of educational content within mhGAP-IG.v2 would be merited.

This research confirms a fundamental difference between China and the overall directive of the mhGAP-IG.v2 (i.e., that primary care plays a key role in diagnosis, psychosocial interventions and initiates antidepressant treatment). Currently, primary care physicians in Shenzhen are not responsible, nor have the skills, for formal diagnosis of depression. Instead, they refer undiagnosed (but suspected) cases to hospital specialists for formal diagnosis and treatment [33,34,53]. Doctors’ desire to invert the mhGAP-IG.v2 flowchart to deal with physical symptoms first is a reminder of the scope of China’s standard medical training which promotes the evidence-based identification of physical causes of disease and is counter-intuitive to a mental health model that determines illness based on subjective mood constructs. Whilst doctors with a Traditional Chinese Medicine (TCM) background naturally consider psychology (or spirituality) as central to health and wellbeing [78], the adoption of a mood-based schema may be rejected by many doctors. Thus, any adaptation of mhGAP-IG.v2 would need to review the evolving role of primary care doctors within China’s wider health system (i.e., consider the merits of extending their diagnosis and treatment responsibilities) and revise the medical curriculum accordingly.

## 5. Strengths and Limitations

This research design successfully paired a capacity building program for the development of primary care provision in Shenzhen with a qualitative research project to better understand doctors’ attitudes to assessing depression in the context of the mhGAP IG.v2. Delegates on the program are considered by their funders to be leaders of healthcare reform and were identified by the research team as potential users of the adapted guidelines. Participants were highly motivated to contribute to the research process and aware of the role they might play in future, locally-led, implementation strategies.

The use of task-based activities gave participants ample time to both explore and reflect upon the mhGAP-IG.v2 content and relevancy to the Community Health Centre (CHC) work-context. The group work setting generated a solution-orientated attitude as group members collaborated to produce a common goal (i.e., an adapted guide). Although research was predominantly bilingual, the dedication of an expert simultaneous translator (experienced in qualitative research practice; fully briefed of study objectives; and aware of the cultural backdrop of both the participants and the primary researcher), ensured both moderator and participants could seek mutual clarification on any issues of uncertainty. Transcripts/translations were also crosschecked against the final findings to ensure fidelity of findings and minimise any misinterpretations.

A limitation of the study is that participants were drawn from an elite group of GPs in senior management roles in Shenzhen’s community health centres who were attending an overseas primary healthcare capacity-building programme. None of the attendees at that time came from the urban district of Nanshan, nor the two rural districts of Pingshan and Dapeng. Whilst nearly all doctors came from different clinics, the study included several doctors from the same CHC in some districts (Guangming, Baoan, Longhua), thus diluting the sample heterogeneity.

Their experiences and views are not necessarily generalizable to all primary care doctors in Shenzhen, or in other parts of China. However, identified by the Shenzhen Health Capacity Building and Continuing Education Centre, as primary care leaders for their communities, they are ideally positioned to understand and describe the day-to-day assessment of depression in their primary care clinics and to provide insightful modifications of mhGAP-Ig.v2 for use in CHCs.

Low levels of mental health literacy has been identified amongst medical personal [25,26,27,28] and research with similar participants, relates that CHC doctors experience of conducting a formal depression diagnosis is limited [33]. Having prior experience of diagnosing depression was not a prerequisite for research participation as this was not the “norm” for general practice at the time of research. Research design attempted to compensate for this by selecting from a leadership cohort and identifying individuals with a self-declared professional interest in mental health as demonstrated by certification or intent to certify in Shenzhen’s relatively new training schemes for primary care mental health.

Finally, research was conducted prior to the COVID-19 outbreak and the clinical presentation landscape for primary care doctors has most likely changed as a result of this significant healthcare event.

## 6. Future Research

Future research would seek to establish a consensus on the applicability and feasibility of proposed modifications to mhGAP-IG.v2 from a much broader group of primary care doctors from across all districts. Additional efforts would need to be directed at Nanshan, which is a key urban, decision-making centre. Alternative research methodologies and sampling strategies would need to be employed to access the remote districts of Pingshan and Dapeng where international collaboration is not yet mainstream.

Ongoing qualitative research should be conducted amongst patients, primary care doctors and the wider mental healthcare team to explore how China’s evolving culture and recent COVID-19 events continue to shape societal attitudes and healthcare response to depressive disorder. In particular, research might explore whether doctors’ conceptualisations of depression have changed and symptom listings need to be altered to take into account psychological risk factors associated with isolation events experienced during the COVID-19 outbreak.

## 7. Conclusions

Primary healthcare doctors regularly see patients throughout the life course of depressive disorder and are ideally placed to be the sentinels of mental wellbeing. Taking account of their cultural frame of reference and unique work context is instrumental to the development of acceptable and appropriate guidelines for use in primary healthcare clinics in Shenzhen. A modified mhGAP-IG.v2 for the assessment of depression would recognize the need for: a culturally sensitive language translation; a broader and less stringent symptom listing; the routine consideration of suicide risk; involving family as part of collaborative, multi-sectorial care; and additional educational content regarding bipolar disorder. Importantly, primary healthcare physicians need to be supported with ongoing mental health training to develop their recognition confidence of depressive disorders and other associated mental disorders such as bipolar disorder.

## Figures and Tables

**Figure 1 ijerph-19-02570-f001:**
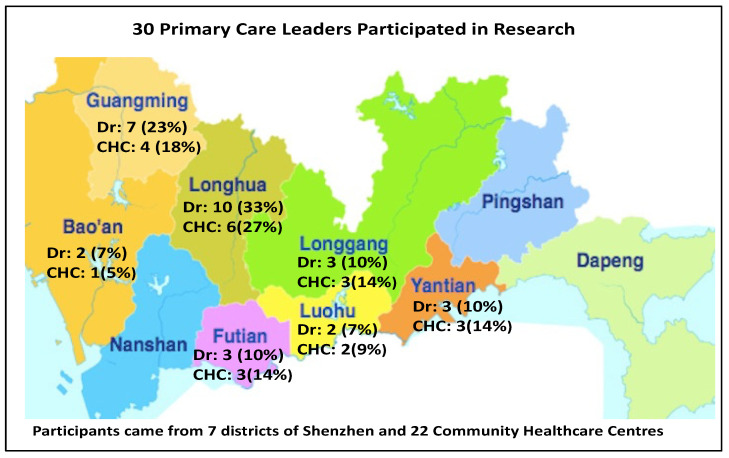
Geographic distribution of participants and community healthcare centres.

**Figure 2 ijerph-19-02570-f002:**
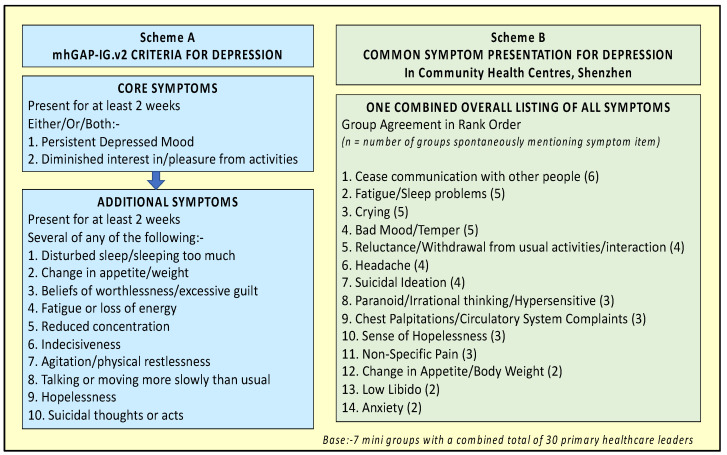
Comparison of mhGAP-IG.v2 criteria for depressive disorder (Scheme A) against Doctor Proposed Listing of Common Symptom Presentation for Depression In Community Health Centres, Shenzhen (Scheme B).

**Table 1 ijerph-19-02570-t001:** Task-based activities employed during workshops to compare participant approach towards the assessment of depression in comparison to mhGAP-IG.v2.

Information Extracted from mhGAP-IG.v2 (Pages 21–25) Presented on a Wall-Chart *	Task-Based Activity Conducted during the Assessment Workshop	Purpose of the Activity
*Common Presentations of Depression*		
Multiple persistent physical symptoms with no clear cause Low energy, fatigue, sleep problems Persistent sadness or depressed mood, anxiety Loss of interest or pleasure in activities that are normally pleasurable	Free Listing: Each mini-group presented with an empty box labelled “Common Presentations of Depression” and invited to draft the contents	To elicit participants’ spontaneous conceptualisation of depression based on professional clinical experience/personal understanding of the condition
*Does the person have depression?*		
Has the person had at least one of the following core symptoms for depression for at least 2 weeks? Has the person had several of the following additional symptoms for at least two weeks? Does the person have considerable difficulty with daily functioning in personal, family, social, educational, occupational, or other areas?	Ranking by Traffic Lights: Each mini-group asked to compare the content and decision-making pathways against their current assessment practice, then record their views on the chart using colour-coded pens (Red: doesn’t apply to their workplace; Orange: uncertain if applies; Green: applicable to workplace; Blue: missing factors)	To identify areas of the guide where the content and sequencing of the decision-making process might need to be adjusted to take into account local, context-specific needs
*Are there other possible explanations for the symptoms?*		
Is this a physical condition that can resemble or exacerbate depression? Is there a history of mania? Has there been a major loss (e.g., bereavement) within the last six months	Label Generation: Each participant invited to record their reflections on adheasive labels and stick them onto the appropriate section of the guide	To explore participants’ awareness and acceptance of the decision-making schema to rule-out alternative diagnoses to consider “depression as likely”.

* Wall-charts were 2m high and located in different parts of the room (one wall-chart/mini-group).

## Data Availability

Data is stored at the University of Melbourne. The data cannot be freely used, as the study is part of a Ph.D. thesis, with the candidate currently working on the remaining data.”

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
