# Peer review of "Listening to the Shenzhen Primary Healthcare Context to Adapt the mhGAP-IG.v2 for the Assessment of Depression: Qualitative Workshops with Primary Healthcare Leaders"

_ijerph, 2022, doi:10.3390/ijerph19052570_

Round 1

Reviewer 1 Report

This is a qualitative study of Chinese primary care clinicians working in community health clinics about their views of screening/identifying patients for depression symptoms.  While a useful and meritorious topic, the primary focus of the study is not clear, and the discussion and conclusions seem to go beyond the data.  

1)In the introduction, the authors neglect to cite any of the large body of work from the US/UK and elsewhere about screening and management of depression in primary care although the purpose seemed to be how to start to implement such practices in China.

2) The authors consistently state that the main role for primary care is to identify 'early stage' depression.  Its not clear what this means.  In fact, patients screened in primary care are identified with both severe and mild as well as long-standing depressive symptoms in part due to stigma and limited access to mental health services in most health care systems around the world.

3) A better description of the recommendations from the mhGAP report would be useful and well as what is the primary aspect they are studying.  The implication is that the focus was on screening items and how they might not be culturally relevant, but the quotes and discussion go way beyond that to speculate about caregiver involvement, a seemingly negative reception among the primary care clinician's that this would be an appropriate role for them, when to engage of mental health specialists, the role of the family, the police etc. What was the focus group guide? (e.g. list of topics to be discussed?)

4) Are the items listed in the Schema box suggested as screening questions by the WHO?  If so, is there a scoring algorithm? Is there an emphasis  from WHO on screening/treatment in primary care? (I suspect so given the lack of specialists in most of the world health care systems.)

5) On p. 2 the authors in the introduction/literature review seem to indicate that primary care is an appropriate place to treat depression and state that physicians can provide psychotherapy and medication.  It is not clear if this is an mgGAP WHO recommendation. In other places MH tx seems to be reserved for psychiatrists and its not clear if they mean primary care in this paragraph (bottom p. 2).  Generally, primary care practitioners do not provide psychotherapy as the authors state, but there are many models of mental health services embedded or closely linked where primary care physicians may provided initial assessment and prescribing of antidepressants, but refer patients for therapy and also refer to psychiatry for treatment resistant or complicated cases.  This whole literature is overlooked.

6) The most powerful part of the manuscript is the reporting of symptoms considered to be depression and how they may differ from the WHO list.  However I see considerable overlap in the two lists and a number of the Chinese-specific items seem to be behavioral manifestations of "persistent depressed mood."  Does this imply merely that wording for screening needs to be adapted?

7) The opening of the discussion focusing on family care giving seems off topic, even though there are some quotes about this.  Again, this suggests that the reader needs to know the framing questions the participants were asked in addition to the symptom list.  Also its a bit strange that there is detailed information about the migrant worker population in this district (which implies many individuals not living with their families) which would seem to make this recommendation not useful for the particular population.

8) The discussion mentions that they interpret the older term 'neurasthenia" as more appropriate for the constellation of somatic symptoms identified by the clinicians.  This should have been teased out better in the results section.  Depression diagnostics generally include both mood and somatic items.  Further they also include level of interference with functioning to assess severity which is not mentioned in the manuscript.

8) The discussion mentions several studies about Chinese adaptations of the PHQ-9 and recommends its use (p, 11)despite quoting a clinician saying that his/her patients didnt understand it.  This needs to be explored more thoroughly.

9) The discussion also talks about anxiety symptoms as part of a depression picture, or a precursor.  While there typically is a strong overlap between anxiety and depression it is not necessarily a precursor. There are accessible, short anxiety screens as well, in use elsewhere and if there is a sense that anxiety is more prevalent, then some discussion should be devoted to that topic and whether it is included in the mhGAP recommendations.

10) The author's emphasize the Chinese emphasis on suicide prevention but nothing is described about how that is screened for and if in primary care. A suicidality question is included in the PHQ screener (although is not present in some more general depressed mood measures) specifically because of the critical need to screen for this problem. It is on the list of symptoms identified by the participants but it is not clear whether this was discussed or whether primary care clinician's in China already have a role in that identification.  

11) Some minor formatting issues:  on p. 3 there are a number of words with randomly inserted '-' marks;  the 'schema' diagram and 'box 1' labels should be changed to Table 1 and Table 2 with appropriate descriptive headings.

Author Response

Thank you for your comments in regards to our manuscript now renamed “Listening to the Shenzhen Primary Healthcare Context to adapt the mhGAP-IG.v2 for the assessment of depression:- qualitative workshops with primary healthcare leaders”. We are grateful for your time and consideration of our work. Please find below our responses to your feedback.

This is a qualitative study of Chinese primary care clinicians working in community health clinics about their views of screening/identifying patients for depression symptoms. While a useful and meritorious topic, the primary focus of the study is not clear, and the discussion and conclusions seem to go beyond the data.

We have clarified the study objective and updated its mention in the introduction section as follows:- “Thus, the objective of this research, was to compare primary care’s assessment approach for depression against the mhGAP-IG.v2 in order to identify context specific modifications that would improve the relevance and acceptability of the guide for use in Shenzhen’s primary healthcare sector.”

We have removed the recommendations box and revised the document throughout to more strongly align with the primary research objective and provide continuity.

1) In the introduction, the authors neglect to cite any of the large body of work from the US/UK and elsewhere about screening and management of depression in primary care although the purpose seemed to be how to start to implement such practices in China.

Our focus was the mhGAP-IG.v2 which provides evidence-based clinical guidance for the assessment of depression. Information in this guide is distilled from these practices that you mention in the UK and USA (as well as other countries across the world) to provide details on “best practise”. Thus we would prefer (for the most part) to refer back to this global standard throughout the manuscript. We have added a paragraph in the background to clarify the guidance provided:-

“mhGAP-IG.v2 is composed of seven chapters (one per priority condition). Clinical guidance for the assessment of depression is provided in the first section of the first module (pg 21-25). It does not assume the format of a screening instrument (i.e. no scoring element included) but instead provides an initial overview of the common presentations of depression, then employs a flow-chart to communicate key decision-making steps for:- i) Does the person have depression? ii) Are there other explanations for the symptoms? Depression is considered if the person experiences at least one core mood-based symptom (i.e. persistent depressed mood and/or diminished interest in or pleasure from activities) AND several other concerns from a symptom listing for at least 2 weeks, AS WELL AS, has difficulty with daily functioning in relationship domains. Depression is only “considered likely”, after ruling-out other physical conditions that might resemble depression (e.g. anaemia, hyperthyroidism, mania and recent bereavement.”   

We do however, refer back to UK guidelines in the discussion based on Ferenchick, E. K.; Ramanuj, P.; Pincus, H. A., Depression in primary care: part 1-screening and diagnosis. BMJ 2019, 365, l794. to explain that general approach to screening.

 “In general, screening is considered helpful (particularly in high risk populations) as a first step in depression diagnosis in health settings with access to reliable screening tools and where resources are available to follow through with an additional diagnostic assessment and provision of access to appropriate treatment [16]

2) The authors consistently state that the main role for primary care is to identify 'early stage' depression. Its not clear what this means. In fact, patients screened in primary care are identified with both severe and mild as well as long-standing depressive symptoms in part due to stigma and limited access to mental health services in most health care systems around the world.

Agreed. We have revised/deleted as appropriate in relevant sections of the manuscript.

3) A better description of the recommendations from the mhGAP report would be useful and well as what is the primary aspect they are studying. The implication is that the focus was on screening items and how they might not be culturally relevant, but the quotes and discussion go way beyond that to speculate about caregiver involvement, a seemingly negative reception among the primary care clinician's that this would be an appropriate role for them, when to engage of mental health specialists, the role of the family, the police etc. What was the focus group guide? (e.g. list of topics to be discussed?)

We have added a table into the methods section "Table 1: Task-based activities employed during workshops to compare participant approach towards the assessment of depression in comparison to mhGAP-IG.v2" and better aligned the results section with following the structure of this table.

We have reworded the recommendation section throughout and clarified where comments are interpretive.

The point about caregiver involvement is that doctors recognise the value of family members in assisting patients to access care (including the initial consultation when an assessment might be made). China’s national policy also places emphasis on the role of family as part of intersectoral care and thus primary healthcare leaders suggest that the guide makes reference to the family’s role to facilitate patient assessments.

Discussion, Paragraph 5, revised to:- “Referencing the role that family members play during the assessment process (and thereafter) within the guidelines could potentially pay respect to, and harness, a viable and cost-effective resource in comprehensive mental health care.”

4) Are the items listed in the Schema box suggested as screening questions by the WHO? If so, is there a scoring algorithm? Is there an emphasis from WHO on screening/treatment in primary care? (I suspect so given the lack of specialists in most of the world health care systems.)

The items listed in the Schema box were extracted from mhGAP-IG.v2. They are not intended to be used as a screener but as clinical guidance for the assessment of depression. There is no scoring algorithm attached. However, as stated in the Results Section, Primary Healthcare Leaders Response to Depression “The assessment section of mhGAP-IG.v2 was developed taking into account the general classification principles of the International Classification of Diseases and Diagnostic and Statistical Manual as available at that time (i.e. ICD-10 and DSM-IV)”.

5) On p. 2 the authors in the introduction/literature review seem to indicate that primary care is an appropriate place to treat depression and state that physicians can provide psychotherapy and medication. It is not clear if this is an mgGAP WHO recommendation. In other places MH tx seems to be reserved for psychiatrists and its not clear if they mean primary care in this paragraph (bottom p. 2). Generally, primary care practitioners do not provide psychotherapy as the authors state, but there are many models of mental health services embedded or closely linked where primary care physicians may provided initial assessment and prescribing of antidepressants, but refer patients for therapy and also refer to psychiatry for treatment resistant or complicated cases. This whole literature is overlooked.

Introduction, Paragraph 5 has been amended to read “In high-income countries depression screening instruments are both available and validated for use at primary care sites [48] and doctors are well placed to provide both psychosocial interventions and drug treatment [49]." This is as per mhGAP-IG.v2 guidance.

Later in Paragraph 5, it has been amended to read “However, in practice, not all CHCs have access to an in-house physician with mental health training, nor are diagnostic tools or depression protocols standardly available, leaving primary care doctors poorly equipped to recognize depression in their communities [33].”

In China, currently it is not legal for primary care doctors to initiate antidepressant treatment, thus their only option to access medication, is to refer patients to specialists in the over-burdened hospital system. We have tried to keep this paper focused on the assessment aspects of mhGAP-IG.v2. We have written a second paper that focuses on the management and follow-up aspects of depression care in the Shenzhen which attempts to tackle the matters that you raise.

6) The most powerful part of the manuscript is the reporting of symptoms considered to be depression and how they may differ from the WHO list. However I see considerable overlap in the two lists and a number of the Chinese-specific items seem to be behavioral manifestations of "persistent depressed mood." Does this imply merely that wording for screening needs to be adapted?

Agree that sensitive translation of behavioural manifestations is needed. We have rewritten Discussion, paragraph 1 to emphasise this point.

“Firstly, this research highlights the importance of using culturally sensitive translations for guideline development. In keeping with previous research, doctors chose to use euphemisms when discussing depression [33]. Their (spontaneous) conceptualisation of depression was devoid of core mood related terminology such as “persistent depressed mood”. Instead, they used expressions such as, “cease communication/talking less” and “fatigue/tiredness”, which are colloquial synonyms for “sadness and depression” and “hurt or despair” respectively [20, 77]. Given the backdrop of intense mental health stigma [24-26, 29-31], patients commonly communicate “depressed mood” in terms of bodily complaints, perceiving these to be a more acceptable premise upon which to seek medical attention [20, 78]. It might be interpreted, that doctors subconsciously take account of the cultural masking of depression and accommodate for this in their choice of language. Thus, an adapted guide would do well to further investigate these social-cultural and linguistic nuances.”

However, other adaptations are also needed. For example, to take account of the sequencing of the decision making process. We have rewritten, Results, Response to Ruling out Alternative Diagnoses, Checking for physical conditions that can resemble or exacerbate depression to improve this aspect:- “The positioning of this question within the flow-chart sequence (i.e. after considering mood-based symptoms) caused considerable debate and prompted doctors to reflect upon primary care’s role within the health system. Some mini-groups wanted to invert the guide to align decision-making with their current day-to-day clinical response to patients….etc.

7) The opening of the discussion focusing on family care giving seems off topic, even though there are some quotes about this. Again, this suggests that the reader needs to know the framing questions the participants were asked in addition to the symptom list. Also its a bit strange that there is detailed information about the migrant worker population in this district (which implies many individuals not living with their families) which would seem to make this recommendation not useful for the particular population.

An additional table in the methods section has been provided to clarify the discussion approach. Table 1: Task-based activities employed during workshops to compare participant approach towards the assessment of depression in comparison to mhGAP-IG.v2) and better aligned the results section with following the structure of this table.

8) The discussion mentions that they interpret the older term 'neurasthenia" as more appropriate for the constellation of somatic symptoms identified by the clinicians. This should have been teased out better in the results section. Depression diagnostics generally include both mood and somatic items. Further they also include level of interference with functioning to assess severity which is not mentioned in the manuscript.

Doctors didn’t overtly mention neurasthenia. This was an interpretative point. Thus we have REMOVED the comment from this section.

9) The discussion mentions several studies about Chinese adaptations of the PHQ-9 and recommends its use (p, 11) despite quoting a clinician saying that his/her patients didnt understand it. This needs to be explored more thoroughly.

We have removed the quote as it did not reflect the majority view.

10) The discussion also talks about anxiety symptoms as part of a depression picture, or a precursor. While there typically is a strong overlap between anxiety and depression it is not necessarily a precursor. There are accessible, short anxiety screens as well, in use elsewhere and if there is a sense that anxiety is more prevalent, then some discussion should be devoted to that topic and whether it is included in the mhGAP recommendations.

Agreed. We have written in the Discussion, Paragraph 3: “Thus any guidance regarding the assessment of depression should also consider anxiety and its prevalence in the community.”

11) The author's emphasize the Chinese emphasis on suicide prevention but nothing is described about how that is screened for and if in primary care. A suicidality question is included in the PHQ screener (although is not present in some more general depressed mood measures) specifically because of the critical need to screen for this problem. It is on the list of symptoms identified by the participants but it is not clear whether this was discussed or whether primary care clinician's in China already have a role in that identification.

As you note, suicidal thoughts/acts are included in the assessment criteria.  Doctors make the point that this item should be moved up the listing to reflect CHC priorities. Although doctors are tasked with improving the identification of suicide, currently they do not use any screening instruments for this. A footnote has been added (corresponding to Discussion, Paragraph 4), to clarify that identification is informal in nature.

12) Some minor formatting issues: on p. 3 there are a number of words with randomly inserted '-' marks; the 'schema' diagram and 'box 1' labels should be changed to Table 1 and Table 2 with appropriate descriptive headings.

Amendments made. Many thanks for identifying these formatting issues. It is greatly appreciated.

Reviewer 2 Report

The aim of the present manuscript was to explore Primary Healthcare Leaders from Shenzhen conceptualisation of depression and their perceptions of World Health Organisation’s (WHO) mental health gap intervention guide (mhGAP-IG.v2.)

Congrats for your work, however, there are a number of issues of concern as outlined below 

 Title: No se Identifying the study as qualitative or indicating the approach (e.g., ethnography, grounded theory) or data collection methods (e.g., interview, focus group)
Abstract: The qualitative methodology used is not mentioned

Methods: It is not clear why you mention that they are "leaders" according to the manuscript, the participants attended a leadership course which does not make them leaders, although it is reported that they had leadership qualities and had 4 years of experience in a practice environment general may not be enough, nor is it clear what is meant by "academic excellence".
The profile of the participants does not seem to be representative of either primary care physicians or a leader.

Results: It is mentioned that doctors from all districts were not included, but it is not clear if the other districts were equally represented.
Strengths and limitations: In the strengths it is even mentioned that they are considered “future” leaders, so they are not leaders at this time, which reinforces the previous comment. It is mentioned that HCC doctors do not have experience in diagnosing depression, it does not remain if this applies to the participants, since they do not have the usual characteristics of HCC, it would be appropriate to identify how confident the participants feel to identify depression or how often their medical practice has made the diagnosis.
Due to the characteristics of the participants, the results could not be extrapolated to HCC or MCT physicians.
Concussion: Conclusions are established that are not actually the product of the research carried out, I consider it should be concluded only based on what was reported by the participants. 

Author Response

Thank you for your comments in regards to our manuscript now renamed “Listening to the Shenzhen Primary Healthcare Context to adapt the mhGAP-IG.v2 for the assessment of depression:- qualitative workshops with primary healthcare leaders”. We are grateful for your time and consideration of our work. Please find below our responses to your feedback.

1)Title: No se Identifying the study as qualitative or indicating the approach (e.g., ethnography, grounded theory) or data collection methods (e.g., interview, focus group)
Abstract: The qualitative methodology used is not mentioned

Title updated as follows:- “Listening to the Shenzhen Primary Healthcare Context to adapt the mhGAP-IG.v2 for the assessment of depression:- qualitative workshops with primary healthcare leaders”

2) Methods: It is not clear why you mention that they are "leaders" according to the manuscript, the participants attended a leadership course which does not make them leaders, although it is reported that they had leadership qualities and had 4 years of experience in a practice environment general may not be enough, nor is it clear what is meant by "academic excellence". The profile of the participants does not seem to be representative of either primary care physicians or a leader.

This section has been revised as follows:

“Inclusion criteria:  identified as a “primary care leader” by the Shenzhen Health Capacity Building and Continuing Education Centre, (the medical regulatory body and vocational training provider for primary care qualification and certification in Shenzhen) as determined by academic excellence (i.e. examination scores for national medical examinations in the top quartile) and assessed during competitive panel interviews to receive a government funded place on The Monash-Shenzhen General Practice Clinical Leadership Training Program [67]; holding senior manager grade or above at a Community Healthcare Centre in any one of Shenzhen’s ten districts; hold clinical responsibility for patients and “see”* depression patients; either hold or intend to study for the primary healthcare mental health certification; English Language proficiency”

At the time of research, many primary health care was relatively new in China. Although on average doctors had 4 years experience working in their current CHC, they generally had 10 years experience as a qualified doctor. Additionally, all doctors were of senior management grade or above.

In terms of general representivity, this is as you note a convenience sample. We have updated the limitations paragraph to emphasise the following:-

“Their experiences and views are not necessarily generalizable to all primary care doctors in Shenzhen, or in other parts of China.”

3) Results: It is mentioned that doctors from all districts were not included, but it is not clear if the other districts were equally represented.

New figure provided to give additional sample details:- "Figure 1: Geographic distribution of participants and community healthcare centres"

Districts were not equally represented. It was a convenience sample.

Strengths and limitations: In the strengths it is even mentioned that they are considered “future” leaders, so they are not leaders at this time, which reinforces the previous comment. It is mentioned that HCC doctors do not have experience in diagnosing depression, it does not remain if this applies to the participants, since they do not have the usual characteristics of HCC, it would be appropriate to identify how confident the participants feel to identify depression or how often their medical practice has made the diagnosis.

They were identified as leaders ”by the Shenzhen Health Capacity Building and Continuing Education Centre." We have removed “future” from strengths and limitations to avoid confusion.

Agreed they do not share the same characteristics of primary care doctors in Western settings because the primary healthcare sector is relatively new in China. It has been previously reported that in general primary care doctors do not feel very confidence regarding mental health conditions. However, we can confirm that they were the appropriate individuals to provide input in this context and as stated in the limitations:-

Research design attempted to compensate for this by selecting from a leadership cohort and identifying individuals with a self-declared professional interest in mental health as demonstrated by certification or intent to certify in Shenzhen’s relatively new training schemes for primary care mental health”.

4) Due to the characteristics of the participants, the results could not be extrapolated to HCC or MCT physicians.
Concussion: Conclusions are established that are not actually the product of the research carried out, I consider it should be concluded only based on what was reported by the participants.

The recommendations box has been removed as this was interpretive. The discussion has been rewritten. Conclusions relate to the revised objectives as stated in the Introduction:-

the objective of this research, was to compare primary care’s assessment approach for depression against the mhGAP-IG.v2 in order to identify context specific modifications that would improve the relevance and acceptability of the guide for use in Shenzhen’s primary healthcare sector.”

Reviewer 3 Report

Dear authors

After reading your manuscript I have the following recommendations:

1.- It would be necessary that reduce the number of words used in your title. Is so extensive, is descriptive, ok, but the excessive extension reduces the interest. I suggest reducing the title extension and converting it into something more attractive to future readers. 

Please, don't forget that the title is the 

2.- Material and methods section

subsection 2.1 you include in your text "XXLocationConcealedXX", why? Is there a situation that allows you to hide the location? 
If so, please explain.

If this was an error, please correct it and include the correct information.

Subsection 2.2. appear, again, "XXConcealedXX" before the title of training programme. why?.

Please explain these situations, because If this was an error, please correct it and include the correct information.

3.-I have not found any reference to the project having been approved by any research ethics committee.
Likewise, I was unable to find information on whether participants signed a consent form to participate in their study.

Please, Could you explain it to me?

4.- In relation to the limitations of the study, I believe it would be appropriate to mention, in addition to the type of sampling, the limitation presented by the type of research carried out - qualitative.

Author Response

Thank you for your comments in regards to our manuscript now renamed “Listening to the Shenzhen Primary Healthcare Context to adapt the mhGAP-IG.v2 for the assessment of depression:- qualitative workshops with primary healthcare leaders”. We are grateful for your time and consideration of our work. Please find below our responses to your feedback.

1) It would be necessary that reduce the number of words used in your title. Is so extensive, is descriptive, ok, but the excessive extension reduces the interest. I suggest reducing the title extension and converting it into something more attractive to future readers.

You will note that we have simplified the manuscript’s title to:- “Listening to the Shenzhen Primary Healthcare Context to adapt the mhGAP-IG.v2 for the assessment of depression:- qualitative workshops with primary healthcare leaders”

2) Please, don't forget that the title is the 2.- Material and methods section

Title updated.

3) subsection 2.1 you include in your text "XXLocationConcealedXX", why? Is there a situation that allows you to hide the location?
If so, please explain.

If this was an error, please correct it and include the correct information.
Subsection 2.2. appear, again, "XXConcealedXX" before the title of training programme. why?.

Please explain these situations, because If this was an error, please correct it and include the correct information.

Apologies for the confusion. The journal requires that we blind reviewers of the location of research and thus these details were removed from your manuscript.

4) I have not found any reference to the project having been approved by any research ethics committee. Likewise, I was unable to find information on whether participants signed a consent form to participate in their study. Please, Could you explain it to me?

Ethical approval was received to conduct this research from The University of Melbourne’s Human Ethics Advisory Group (HEAG), June 2018 (ID:1851783.1). Apologies this was obscured from your manuscript for blinding purposes.

5).- In relation to the limitations of the study, I believe it would be appropriate to mention, in addition to the type of sampling, the limitation presented by the type of research carried out - qualitative.

The limitation section has been expanded to include the key limitation of qualitative research (i.e. non-representative). “The participants in this study were drawn from an elite group of GPs in senior management roles in Shenzhen’s community health centres who were attending an overseas primary healthcare capacity-building programme. None of the attendees at that time came from the urban district of Nanshan, nor the two rural districts of Pingshan and Dapeng. Whilst nearly all doctors came from different clinics, the study included several doctors from the same CHC in some districts (Guangming, Baoan, Longhua), thus diluting the sample heterogeneity. Their experiences and views are not necessarily generalizable to all primary care doctors in Shenzhen, or in other parts of China. However, identified by the Shenzhen Health Capacity Building and Continuing Education Centre, as primary care leaders for their communities, they are ideally positioned to understand and describe the day-to-day assessment of depression in their primary care clinics and to provide insightful modifications of mhGAP-Ig.v2 for use in CHCs.”

Round 2

Reviewer 1 Report

Authors have substantially improved the manuscript and clarified the rationale and suggestions for follow-up concerning how to adapt depression screening to the Chinese primary care context.  Some of the points raised such as family involvement, understanding of bipolar, and the interaction with specialty care remain slightly out of the main focus of evaluating the WHO guide for assessment of depression in primary care, but those are minor concerns.  It might have been helpful for the authors to distinguish themes such as these separately from the main focus as proposed, but not major enough to require more rewriting.  I think readers will gain an overall understanding of the state of primary care clinician involvement in and barriers to in identifying mental health conditions in China, and gain an appreciation of the major work that would be required in clinician training and system redesign, given the way current primary care training and scope of practice is implemented.  

Reviewer 3 Report

.